# GraphDDP: a graph-embedding approach to detect differentiation pathways in single-cell-data using prior class knowledge

Fabrizio Costa[1,2], Dominic Grün[3] & Rolf Backofen [2,4]

Cell types can be characterized by expression profiles derived from single-cell RNA-seq. Subpopulations are identified via clustering, yielding intuitive outcomes that can be validated by marker genes. Clustering, however, implies a discretization that cannot capture the continuous nature of differentiation processes. One could give up the detection of subpopulations and directly estimate the differentiation process from cell profiles. A combination of both types of information, however, is preferable. Crucially, clusters can serve as anchor points of differentiation trajectories. Here we present GraphDDP, which integrates both viewpoints in an intuitive visualization. GraphDDP starts from a user-defined cluster assignment and then uses a force-based graph layout approach on two types of carefully constructed edges: one emphasizing cluster membership, the other, based on density gradients, emphasizing differentiation trajectories. We show on intestinal epithelial cells and myeloid progenitor data that GraphDDP allows the identification of differentiation pathways that cannot be easily detected by other approaches.

[1] Department of Computer Science, University of Exeter, Exeter EX4 4QF, UK. [2] Department of Computer Science, Albert-Ludwigs-Universitaet, 79110 Freiburg, Germany. [3] Max Planck Institute of Immunobiology and Epigenetics, Freiburg 79108, Germany. [4] Centre for Biological Signalling Studies (BIOSS), University of Freiburg, Freiburg 79104, Germany. These authors contributed equally: Dominic Grün, Rolf Backofen. Correspondence and requests for materials should be addressed to D.Gün. (email: gruen@ie-freiburg.mpg.de) or to R.B. (email: backofen@informatik.uni-freiburg.de)

One of the most important tasks in single-cell RNA-seq is to identify cell types and functions from the generated transcriptome profiles. State-of-the-art approaches for cell type classification use clustering to identify subpopulations of cells that share similar transcriptional profiles (e.g.[1-4], see[5,6] for recent reviews). The development of tailored clustering approaches, including measurements for the similarity of transcriptome profiles, is complex and subject to active research[4,7-12]. While this line of research is very successful in determining main cell types, the clustering hypothesis implies a discretization that does not reflect the nature of differentiation as a continuous process. This is especially true for rare cell types such as stem cells. One possible solution is to give up at the detection of subpopulations and cell identities altogether. Examples are Monocle[13], which determines a pseudo-time associated with differentiation progress from the similarities between cell profiles, the use of diffusion maps to directly determine differentiation trajectories[14], or graph-based approaches like Wishbone[15]. However, it would be much more useful to combine clustering with differentiation pathway visualization since the clustering of major cell types can serve as an excellent validation tool. In particular, clusters frequently represent metastable intermediate differentiation stages or stable end points, respectively, and can thus serve as anchor points, facilitating the derivation of differentiation trajectories.

The million dollar question therefore is how to integrate both views in the most efficient way. Current approaches visualize the cell types using dimensionality reduction techniques like principal component analysis (PCA), multi dimensional scaling (MDS) or t-distributed stochastic neighbor embedding (t-SNE)[16], which allow the easy detection of instances (cells) that are distant from cluster centers, thus pointing to possible differentiation pathways. There are two issues with this strategy. First, each dimensionality reduction technique has a specific bias that determines which type of information is preserved in the reduction. The PCA embedding identifies the two orthogonal axis along which data exhibits maximal variance which corresponds roughly to the two main directions of change; when there are multiple factors influencing data variability, a two dimensional PCA ends up explain only a small fraction of the total variance in the data and hence does not offer a clear separation for each factor. MDS is mainly constrained by the global arrangement and can end up distorting the local arrangement. The popular t-SNE depends on a scaling parameter (called perplexity) which, if not set correctly, yields a layout with data points segregated in several detached groups positioned arbitrarily relative to each other. Furthermore, outliers corresponding to rare cells can be grouped together solely due to their dissimilarity to abundant groups. Second, and more importantly, the classical dimensionality reduction approaches are unsupervised, e.g. they do not take into account class information available, for example, from a prior clustering phase. The recent StemID algorithm[17], which utilizes cluster medoids as anchor points, is a first attempt of combining cluster information and trajectory inference. However, this algorithm still applies t-SNE for visualization of the results.

## Results

**The GraphDDP layout approach**. To overcome the above mentioned limitations, we developed GraphDDP (for **G**raph-based **D**etection of **D**ifferentiation **P**athways), a visualization approach that exploits prior information, provided as a user defined clustering assignment, to detect differentiation pathways. When displaying single-cell data there are multiple criteria that need to be optimized at the same time. On the one hand, instances belonging to the same class should be visualized as a compact (often convex) region. On the other hand, we want to

visually identify differentiation pathways. In this case we would prefer a more distributed layout of the cells, where differentiating cells are placed in the vicinity of the most strongly related class with intermediate cases transitioning between the groups representing the different cell states.

In order to find a visualization of the data capable to integrate both aspects, we employ a force-based layout approach (see Fig. 1 and Methods). Our pipeline determines at first the pairwise similarities between all cells (based for example of their expression profiles). These similarities are converted into a preferred distance between cells (more similar cells should be closer to each other) in the layout algorithm. We use two different types of edges to account for the two competing visualization criteria: (a) to emphasize class membership we add edges connecting each instance (i.e. cell) to its k-nearest neighbors from the same class; (b) to detect possible differentiation pathways we also add edges from an instance to the densest neighbors of a different class. These are called shift-edges as they implement the mode-seeking approach of the quick shift algorithm[18]. The key idea here is that a differentiation pathway should connect regions of high point density. There are two effects that contribute to the detection of differentiation pathways from density information. The first effect is the confluent differentiation of cells along the same pathway. For example, suppose there is a differentiation pathway from a class A to a class B. Then for each instance of A that differentiates to B, the nearest neighbors of this instance in A should be more dense in B than in any other class C due to confluent differentiation. The second effect is the tendency of progenitor cells to be on average more similar to their descendant mature cells than mature cells of distinct classes are with respect to each other. Therefore, differentiation pathways follow the gradient of densities from higher density of progenitors towards lower densities of mature cell types. To improve the "visual contrast" we introduce the concept of "confidence strength" for the user-provided class annotation. The confidence is used to trade-off the importance of the two different types of information. A strong confidence score contracts the desired distance between elements of the same class. Note that the layout is stable over a whole range of possible confidence values (see Supplementary Fig. 1).

To investigate the usefulness of our approach, we applied it to different datasets with user-supplied clustering. First, we re-analyzed a recently published single-cell transcriptome dataset[17] of intestinal epithelial cells. This cell population comprises Lgr5-positive intestinal stem cells and their descendants of all five lineages, i.e., absorptive enterocytes, mucus-secreting goblet cells, anti-microbial Paneth cells, diverse sub-types of hormone-secreting enteroendocrine cells, and tuft cells[19,20]. To analyze the dataset we followed the procedure used by[17] and selected 462 cells. The clustering analysis presented in the original study successfully discriminated mature cell types of distinct lineages and revealed additional clusters representing intermediate differentiation stages (Fig. 2a) and the lineage tree inferred by StemID was in agreement with the current model of intestinal differentiation (Fig. 2b). However, the t-SNE map representation did not reveal these differentiation trajectories, since cells of distinct lineages ended up in detached groups (Fig. 2a). In contrast, our GraphDDP algorithm positioned the stem cell cluster 7 in the center and assembled the clusters of distinct lineages onto connected differentiation trajectories. For instance, the GraphDDP analysis revealed cluster 1 as a common progenitor of Paneth and goblet cells, which emerge from a common trajectory. Moreover, our visualization suggests distinct branches of enteroendocrine cells, with cells in cluster 15 giving rise to Cck+ (cluster 12) and Sst+ subtypes (cluster 9), and cells in cluster 16 being progenitors of Gip+ (clusters 3, 24, and 26)

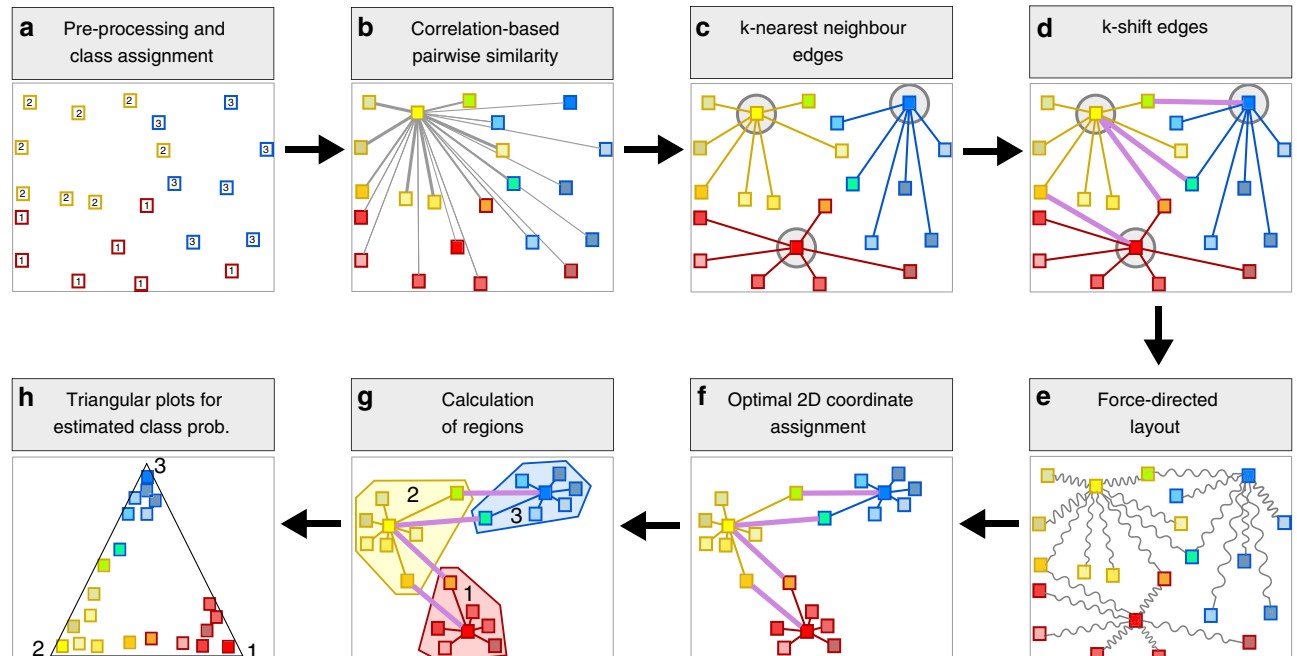

**Fig. 1** Steps in our visualization approach. **a** Each cell is initially assigned to the class as determined by the user-provided clustering; furthermore, additional pre-processing such as filtering and feature selection is done. **b** For each pair of cells the similarity of the expression profiles is calculated using different metrics (see Methods). **c** To emphasize class membership in layout, we add for each cell an edge to the k-nearest neighbors of the same class; each edge is annotated with the desired distance between the two cells. **d** To visualize differentiation pathways, we add another type of edge called k-shift-edges, which connects cells to the $k'$ densest neighbors of a different class. **e** A force layout algorithm interprets each edge as a spring. **f** The optimal 2D configuration is determined minimizing the total energy of the systems. **g** We determine the convex hull of a given class in the layout. **h** Ternary plots are provided to further investigate differentiation pathways. Using a multi-class prediction approach, cells that are clearly members of a class are close to the corners, cells on the differentiation pathway between two classes lie on the corresponding edges, and undetermined ones are placed in the center of the plot

and *Tac1+* (cluster 11) subtypes. Enterocytes emerge from transit amplifying cells in cluster 5, consistent with the StemID analysis. Thus, visual representation by GraphDDP successfully captures the differentiation trajectories obtained by StemID prior to dimensional reduction and reveals additional lineage-specific progenitor states.

A larger scRNA-seq data set was published by the Amit group[21], aiming at resolving heterogeneity across bone marrow resident myeloid progenitors. They identified 19 classes by clustering, which were then grouped together to highlight a developmental continuum. We have analyzed this data using standard visualization tools like PCA, MDS and t-SNE. However, all these approaches fail to indicate any differentiation pathways (see Supplementary Fig. 2). Using our GraphDDP approach with a strong confidence score, we could clearly detect differentiation pathways (see Fig. 3). A subset of these classes (in the following called meta-cluster) that was found both by our approach and in the original paper consists of C7-C1, which represents erythrocyte differentiation. The cells in this meta-cluster show a strong expression of some known erythrocyte transcription factors (TFs) such as *Klf1*, *Gata1* and *Gfi1b*. Concerning differentiation pathways, C7 is clearly an early erythrocyte progenitor and C1 the end-point of differentiation marked by strong expression of hemoglobin. However, as can be seen in Fig. 3, we found indications that there are alternative possible pathways within this meta-cluster. Here, especially the relation of C2, C3, and C4 seems to be unresolved, while the differentiation order was resolved for C7, C6 and C5. This is reflected by expression profiles of different markers for erythrocyte differentiation. *Car1* and *Add2* support a differentiation ordering C4→C3→C2→C1, whereas the expression profile of other proteins related to erythrocyte progenitors like *Ermap* and *Add1* indicate a different

order, namely C4→C2→C3→C1. In order to investigate this in more detail, we developed a more comprehensive approach to visualize pathways. Given the user-defined classes, we tackle the visualization task using a multi-class classification approach. Here, we estimate the class probability for each cell using logistic regression on the cell profiles. In order to inspect whether a class is a transitional state between two other classes, we use a ternary plot of the estimated class probabilities for all cells of the putative transition classes. In a ternary plot, each vertex represents one of the three classes. For any point in the plot, the probability for each of the three classes is given by the distance to the edge located opposite to the class vertex. To investigate for example whether C6 is a transition state between C7 and C5, we estimate the probability of the cells of C5, C6, and C7 to be a member of these classes. As shown in the ternary plot for C5, C6, and C7 in Fig. 3, the cells show a clear tendency to be either towards the line between C6 and C5, or are close to the line between C6 and C7. If a cell is close to the line C6 and C5, this implies that the probability for C7 is negligible, and the cell is on the C6→C5 pathway. We have also introduced a confusion score (see Methods), which has a value between 0 and 1. For the C5-C6-C7 subgroup the value is 0.06 and thus close to 0, which indicates a clear pathways. As a counter example with a confusion score of 0.72, we do not see such a trend for the cells in C2, C3, and C4. These cells have very often similar probabilities for C2, C3, and C4, and thus are placed predominantly in the center of the triangle. This implies that there is no preference for a specific differentiation pathway in these three classes.

The other prominent meta-cluster C13→C16 consists of neutrophil cells and progenitors. Here, the differentiation pathway is even clearer than in the erythrocyte meta-cluster, as we can see from the edges indicating differentiation trajectories between

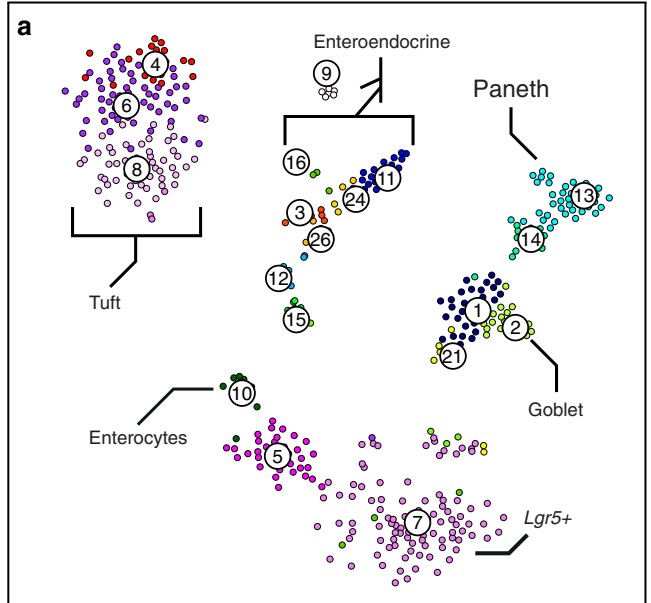
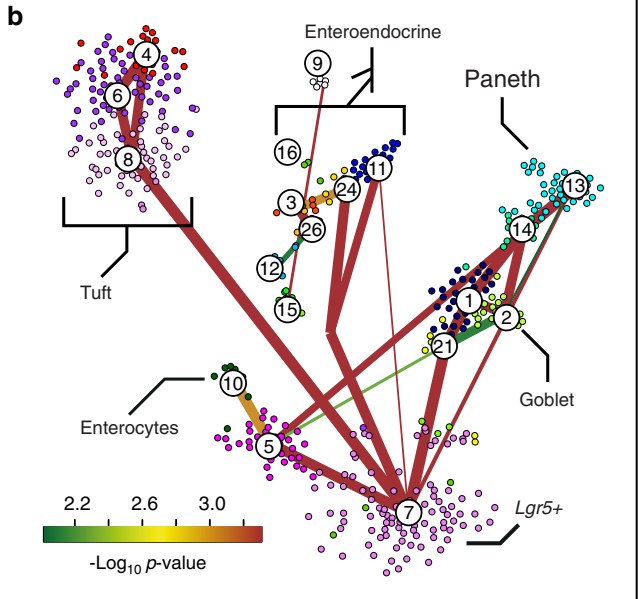
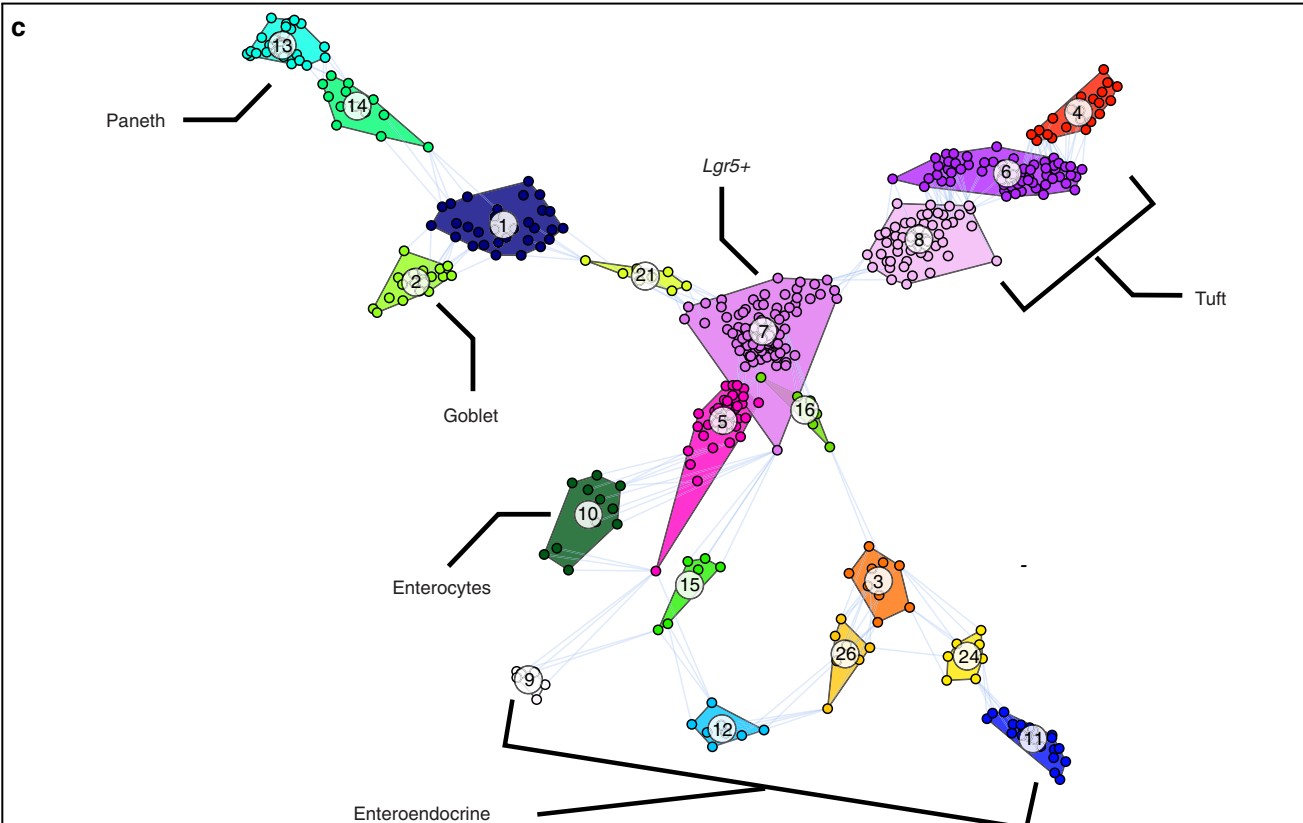

**Fig. 2** GraphDDP reveals differentiation trajectories of intestinal epithelial cells. **a** t-SNE map representation of intestinal epithelial single-cell transcriptome data from Grün et al.[17]. Clusters, highlighted in different colors, were derived by RaceID2 in the original study and correspond to distinct cell types or progenitor stages. **b** The lineage tree inferred by StemID is overlaid on cell clusters. Thicker links reflect higher coverage of a link by cells, and the color reflects the significance of a link measured by a logarithmic p-value. The lineage tree was found to be in good agreement with the current model of intestinal cell differentiation. **c** Visualization of the intestinal epithelial data by GraphDDP. The convex hull of each cluster is shown and shift edges are depicted to reflect the relations between clusters. The representation places stem cells in the center, recapitulates the differentiation trajectories shown in **b** and identifies novel lineage-specific progenitor relations (see text)

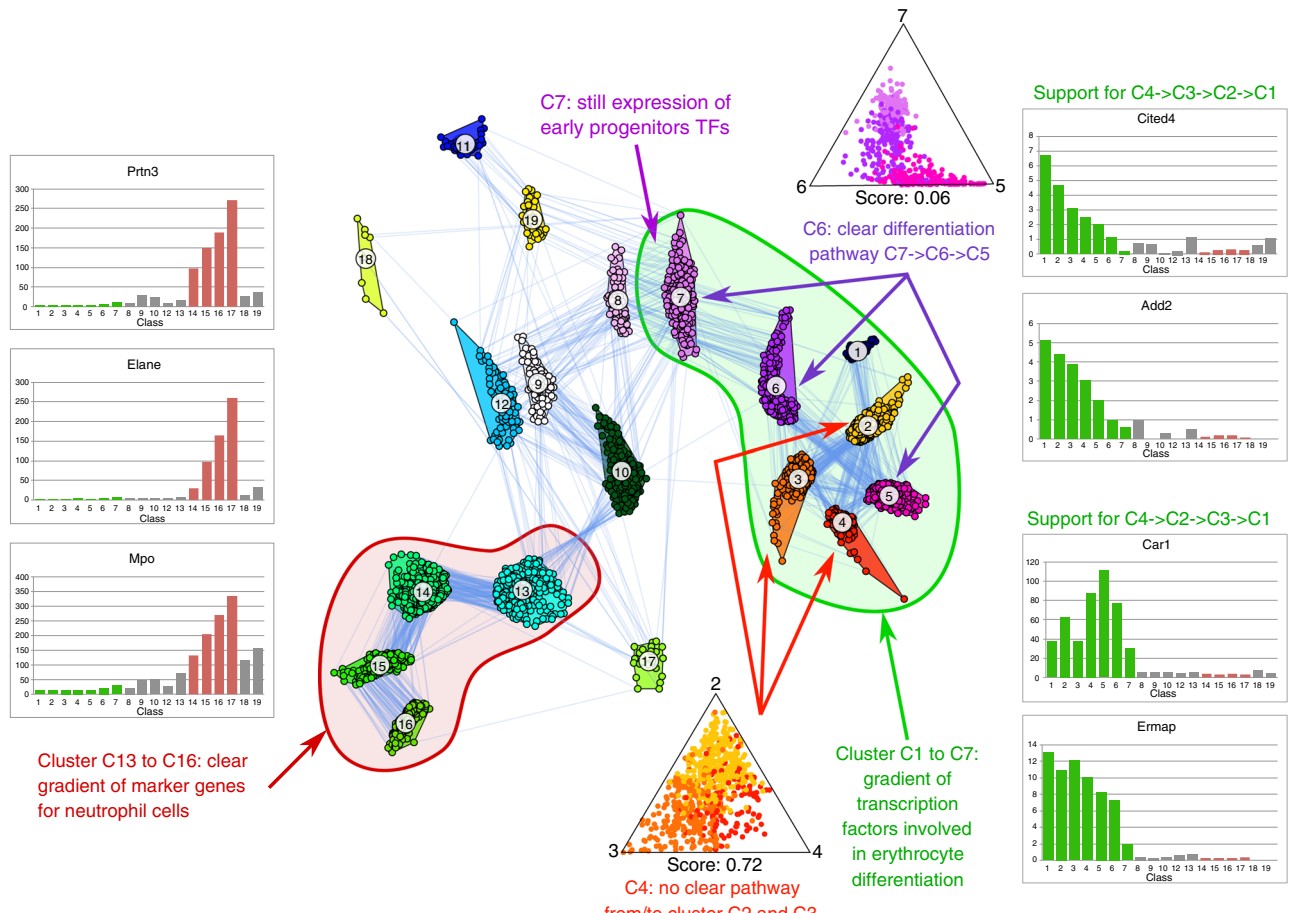

**Fig. 3** Visualization of scRNA-seq data[21] of bone marrow resident myeloid progenitors. The edges represent the denser neighbor of different class (i.e., k-shift edges), indicating differentiation trajectories. The meta-cluster consisting of C1–C7 (encircled in green) represents erythrocyte differentiation, with C1 being the endpoint expressing hemoglobin and C7 being an early erythrocyte progenitor. The differentiation order is clear for clusters C7, C6, and C5, as indicated by many k-shift edges between C7, C6 and C6, C7 in the layout. This is also supported by the ternary plot for C7, C6, C5 (upper triangle), where the cells are mostly located close to the C7–C6 line, or to the C6–C5 line of the triangle. The associated confusion score (see Methods) of 0.06 also clearly indicates a transition. For C2, C3, and C4, there is no obvious ordering, as they are connected by many k-shift edges between all pairs of combination (C2–C3, C2–C4, and C3–C4). Again this is supported by the ternary plot (lower triangle), with many cells in the middle of the triangle, indicating an equal likelihood to be classified as being a member of C2, C3 or C4. The associated high confusion score (0.72) also clearly indicates that there is no clear transition. This divergence can also be seen when looking at the expression profile of different markers for erythrocyte differentiation (plots on the right), which support the order C4→C2→C3→C1, as well as C4→C3→C2→C1. The meta-cluster consisting of clusters C13, C14, C15, and C16 (encircled in red) on the other hand relates to neutrophil differentiation, which is supported by the expression profiles of marker genes for neutrophil cells (plots on the left)

successive classes. This is also supported by expression profiles of neutrophil markers such as *Prnt3*, *Elane*, and *Mpo*. We can observe a pronounced expression, as well as a gradient for these markers in C13 to C16 (see Fig. 3).

**Comparison to other approaches**. We proposed an approach that makes use of prior clustering information and that uses two types of edges to obtain a layout that can at the same time visualize compact groups and transitions between groups. We have compared the quality of the layout with that obtained by t-SNE with a higher perplexity parameter in order to avoid the formation of disconnected components. As shown in Supplementary Fig. 4, although some additional pathways can be detected, a clear class separation is lost.

We then investigated the effect and relative importance of the two types of edges, namely the k-NN edges and k-shift edges. A recently published method called SPRING[22] uses a force-directed layout for a k-nearest-neighbor graph to visualize scRNA-seq

data. To mitigate the influence of a different data preprocessing, we compare to the SPRING method using our code implementation. We have tried three different approaches. First we used a small number of k-NN edges ($k = 3$) without resorting to any confidence contraction. Supplementary Fig. 5 shows that this yields disconnected layouts with overlapping clusters. Second, we tried a stronger neighborhood ($k = 7$). As shown in Supplementary Fig. 6a), this yields a noisy layout with fragmented classes. In Supplementary Fig. 6b), we added prior information to contract edges that are within one class (note that this is an extension of the pure k-NN-based approach used in SPRING). In this case the layout seems to improve, but it is still noisy, especially for less populated classes.

Finally, we assessed whether pseudotime-based approaches are able to identify pathway information. We used TSCAN[23] for our datasets (see Supplementary Fig. 7a–d). For the intestine data set, TSCAN accepts maximal 6 clusters (we provided the actual cluster number as parameter) and displays only one pathway (sub-figure a). When displaying the pseudotime on our layout, we

find that only the major pathway (roughly from left to right in our layout) can be detected and that all other pathways are lost (sub-figure b). When we superimpose the classes detected by TSCAN on the GraphDPP layout (sub-figures c+d) we do not see a good correspondence between the class notion and the visual grouping.

For the myeloid data (see Supplementary Fig. 8), we have used the same number of classes given to GraphDPP for TSCAN (i.e. 20). Once again TSCAN can detect only one linear pathway, compatible with neutrophil cells, early progenitor and erythrocyte differentiation (sub-figure a). However, when we visualize TSCAN pseudotime, we observe (1) large differences in pseudotime within individual clusters, (2) we cannot clearly see major differentiation pathways (sub-figure b) and (3) different groups tend to share the same pseudotime. This class splitting can be clearly seen when comparing the class labels given by the original publications (sub-figure c) and the TSCAN defined classes (sub-figure d).

Finally, we validated whether GraphDPP can retrieve known biological differentiation pathways where ground truth data exists. In mouse, hematopoietic stem cells (HSCs) give rise to a multipotent progenitor population, which shows early segregation into sub-populations biased towards distinct lineages. Megakaryocytes branch-off first[24] followed by a segregation into erythrocyte progenitors, on the one hand, and lymphoid-primed multipotent progenitors (LMPP), on the other hand[25]. The latter further differentiate into common lymphoid progenitors (CLP) of natural killer cells as well as, predominantly, B cells, and granulocyte-macrophage progenitors (GMP). Using a recently published large single-cell RNA-seq dataset of mouse multipotent hematopoietic progenitors[26], application of GraphDDP on clusters obtained with RaceID3[27] reflects this lineage tree architecture clearly, with HSCs in the center, proximal separate branches of erythrocytes and megakaryocytes, and an LMPP population upstream of B cell progenitors and progenitors of neutrophils and monocytes (see Supplementary Fig. 9). Remarkably, GraphDDP layout separates megakaryocyte progenitors from HSCs and B cells although they were initially clustered together. Similarly, the layout visually separates B cell progenitors from naïve LMPPs which were also co-clustering. In contrast, Monocle2[28], a state-of-the-art method for lineage tree inference, did not correctly resolve this hierarchy. Although it recovered major branches, CLPs were assigned to a separate branch from B cells. Furthermore, GMPs appeared as an outgroup, while the tree structure suggests a common upstream progenitor of B cells and erythrocytes. Hence, GraphDDP shows excellent performance in recovering actual biology for a well-characterized ground-truth dataset superior to established methods.

## Discussion

We have presented GraphDDP, an informed visualization approach for the detection of differentiation pathways in single-cell transcriptome data. There are competing aspects for the analysis of scRNA-seq data, namely the detection of major cell-types and the investigation of differentiation trajectories. As the clustering of single-cell transcriptome data for the detection of cell types is a very active field of research and provides an intuitive way to analyze and validate this type of data, we developed a novel approach that combines both analysis aspects. We found that GraphDDP could detect differentiation trajectories in intestinal epithelial and myeloid progenitor cells that could not be identified with state-of-the-art visualization approaches such as t-SNE, PCA or MDS. We furthermore compared our approach to recent visualization approaches, including pseudotime-based methods like TSCAN and Monocle2, as well as k-nearest neighbor approaches like SPRING.

We found that GraphDPP could correctly visualize the hierarchy of cells and associated pathways that were not determined by the other approaches. Thus, we expect that GraphDDP can reveal novel insights in differentiation dynamics of a variety of cell populations profiled by single-cell RNA-seq.

## Methods

**Overview**. We propose an embedding method based on a graph layout technique that relies on an external class assignment. Each cell is modeled as a node in a graph and edges are created only between selected pairs of cells. Problem specific priors can be introduced in the algorithm by choosing which edges to instantiate. The 2D embedding is then computed using a standard force directed graph layout technique.

**Pre-processing**. We represent the $n$ cells, or instances, in a vector space $\mathbb{R}^g$, where $g$ is the number of genes. Initially feature values correspond to the raw number of reads in each cell associated to a specific gene. We considered several pre-processing steps (see Fig. 1a, b): (1) class filtering, (2) feature normalization, (3) feature selection, and (4) feature transformation. These pre-processing steps effectively improve the detection of differentiation pathways. Omitting some if these steps usually leads to a layout with less well defined grouping of clusters into differentiation trajectories (see Supplementary Fig. 3).

Class filtering is performed to remove the clusters that contain fewer cells than a user defined threshold. Feature normalization equalizes the cumulative number of reads per cell, i.e., the actual counts are divided by total number of UMIs per cell.

Feature selection is the removal of non-informative gene expression counts. Here we use the recursive feature elimination[29] (RFE) method. This approach is based on a selected estimator that can assign weights to features, such as the coefficients of a linear model. RFE recursively considers an increasingly smaller set of features. RFE is initialized with as estimator trained using all the available features. Then, the 20 features with the smallest absolute weights are removed. This step is recursively repeated until the estimated accuracy score computed via a 5-cross-validation is increasing, i.e., until the removal of the features actually improves the predictive capacity. The estimator on which the RFE procedure is based is a linear support vector machine (SVM) with stochastic gradient descent[30] (SGD) learning. In the SGD, the gradient of the hinge loss is computed at each sample and the model is iteratively updated using a decreasing learning rate. As it is standard in SVM, we use the squared euclidean norm $L_2$ of the model parameter vector as a regularizer to shrink the parameters towards the zero vector. The multi-class problem is solved using the "One Versus All" (OVA) strategy, where only one classifier per class is used and for each classifier, a single class is fitted against all the other classes. For all other parameters we use the defaults of the SVM SGD implementation provided by scikit-learn[31].

Feature transformation builds a different feature representation on the basis of the original protein expression counts. We found the transformation in the Pearson product-moment correlation coefficient matrix to be a useful data transformation as it allows to work only on the shape of the expression pattern (see Fig. 3 in Supplementary Material for an example of what are the effects of not applying the transformation) and has superior performance in revealing differentiation trajectories compared to alternative metrics for various datasets. This transformation maps instances to vectors in $\mathbb{R}^n$, employing as descriptive features the correlation in expression of this instance with respect to all available instances in the data set. Given the vector representation of a cell profile $x \in \mathbb{R}^g$ we consider the sequence of all the cell profiles $\{z_i\}_{i=0}^{n-1}$ and compute the $i$th entry in the new representation $p \in \mathbb{R}^n$ as $p[i] = \rho_{x,z_i} = \frac{\text{cov}(x,z_i)}{\sigma_x \sigma_{z_i}}$ where the covariance is defined as

$$\text{cov}(x, z) = E[(x - \mu_x)(z - \mu_z)]$$ and the standard deviation is $\sigma = \sqrt{E[(x - \mu)^2]}$ where $E$ is the expectation and $\mu_x$ is a compact form to indicate $E[x]$.

**Modeling priors via edges definition**. In order to detect differentiation pathways we would like (a) cells belonging to the same class to be positioned near each other, but at the same time we want to (b) identify transitions across different but related classes. For each specific case we introduce a different type of edge in the graph (see Fig. 1c, d). We will use the concepts of distances to define neighbors and density gradients to define transitions between classes. In order to bias the layout solution to encourage (a), we connect each instance to its k-nearest neighbors if they have the same class. Given a pair of instances $p_i, p_j \in \mathbb{R}^n$ we then use the Euclidean distance $\|p_i - p_j\|^2$ to compute the k-nearest neighbors. The bias for (b) is enforced by connecting each instance to its $k'$ nearest denser neighbors if they have a different class. More in details: given an instance $p_i$, we consider its closest neighbors, and take the $k'$ nearest that have a higher density and a different class than $p_i$. We call these cells k-shift neighbors to indicate a similar mode seeking behavior as done in the quick shift algorithm[18]. In order to enforce the principle of locality (we do not want to link instances that are at the opposite sides of the embedding space) we further constraint the k-shift edges to be within a user specified threshold value $h$ that we call the *knn horizon*: the shift links are defined as the $k'$ nearest neighbors with a higher density and a different class that are also among the $h$ nearest neighbors.

We chose denser neighbors rather than just any neighbor (of a different class) in order to identify possible differentiation paths: if a cell type $A$ evolves in a more differentiated cell types $B$, as explained above, this is likely to follow a density gradient from dense progenitor to less dense mature cell types. Furthermore, confluent differentiation would materialize in connections between points of high densities.

While density estimator exist that are based on the notion of distances, for example the average pairwise distances, we note that these approaches are quite sensitive to outliers. To gain robustness, we exploit the normalization property of the cosine similarity and define density $D(p_i)$ of an instance $p_i$ as the average pairwise cosine similarity

$$D(p_i) = \frac{1}{n} \sum_{j}^{n} \frac{\langle p_i, p_j \rangle}{\sqrt{\langle p_i, p_i \rangle \langle p_j, p_j \rangle}}.$$

**Layout algorithm.** The procedure we described yields an unweighted instance graph $G = (V, E)$ where $V$ is the set of vertices (one per cell profile) and $E$ is the set of edges or selected pairwise relations between cell profiles. Before the layout phase we assign a desired length to each edge proportional to a desired distance between its endpoints $i$ and $j$. We then use a force directed layout algorithm to determine the optimal position of each vertex (see Fig. 1e, f). Here we use the algorithm from [32] (KK) to obtain a layout that can accommodate the specified edge lengths.

In detail, we model our data set as a dynamic system of particles, with coordinates $v_i = (x_i, y_i) \in \mathbb{R}^2$, mutually connected to each other by springs of strength $k_{ij}$. We seek the layout that minimizes the total energy of the system:

$$E = \sum_{i=1}^{n-1} \sum_{j=i+1}^{n} \frac{1}{2} k_{ij} \left( \left| v_i - v_j \right| - d_{ij} \right)^2$$

where $d_{ij}$ is the desired distance between $i$ and $j$ defined as the length of the shortest path between $i$ and $j$ on $G$ (which can be readily computed using the Dijkstra's algorithm). The spring strength is defined as $k_{ij} = K/d_{ij}^2$ with constant $K$. The computation of the optimal layout is based on the observation that the necessary conditions for a local minimum are that $\frac{\partial E}{\partial x_m} = \frac{\partial E}{\partial y_m} = 0$ for $1 \le m \le n$. However this yields $2n$ simultaneous non-linear equations that are not independent. The authors suggest to adopt an iterative approximation scheme where only one particle $m$ at a time is moved to its stable point while all other particles positions are frozen. This allows to use a two dimensional Newton-Raphson method to minimize $E$ as a function of only $x_m, y_m$. Each time the particle with the largest contribution to the energy of the system is chosen, i.e. $\arg \max_m \Delta_m$ where $\Delta_m = \sqrt{\left( \frac{\partial E}{\partial x_m} \right)^2 + \left( \frac{\partial E}{\partial y_m} \right)^2}$.

The starting position is not critical and all particles are hence initially placed on a regular $n$-polygon. Each selected particle $m$ iteratively updates its position $x_m, y_m$ as: $x_m^{(t+1)} = x_m^{(t)} + \delta x$, $y_m^{(t+1)} = y_m^{(t)} + \delta y$ for $t = 0, 1, \ldots$ where the increments are computed by satisfying the following linear equations:

$$\frac{\partial^2 E}{\partial x_m^2} \left( x_m^{(t)}, y_m^{(t)} \right) \delta x + \frac{\partial^2 E}{\partial x_m \partial y_m} \left( x_m^{(t)}, y_m^{(t)} \right) \delta y = -\frac{\partial E}{\partial x_m} \left( x_m^{(t)}, y_m^{(t)} \right)$$

$$\frac{\partial^2 E}{\partial y_m \partial x_m} \left( x_m^{(t)}, y_m^{(t)} \right) \delta x + \frac{\partial^2 E}{\partial y_m^2} \left( x_m^{(t)}, y_m^{(t)} \right) \delta y = -\frac{\partial E}{\partial y_m} \left( x_m^{(t)}, y_m^{(t)} \right)$$

The coefficients for these equations can be computed in close form starting from the total energy of the system and taking the required derivatives with respect to $x_m$ and $y_m$. The algorithm terminates when $\Delta_m$ becomes sufficiently small.

**User defined confidence strength in class assignment.** In order to obtain more pleasing and informative layouts, the user can decide to enhance the perceived separation of classes by increasing the strength of the constraint (a) i.e. the k-nearest neighbors edges over (b) i.e. the k-shift edges. In this way the user can ensure that cells belonging to the same class are positioned near each other notwithstanding the global relationships between different classes imposed by the k-shift edges. Please note that the user-defined clustering is a critical prior information, and wrong layouts are generated if wrong or low quality class assignments are provided. In more detail, we encode the user confidence on the class assignment as a real positive value $C$. Finally, the k-nearest neighbors edges are favored over k-shift edges by contracting their desired distance to $d(x, z)/(1 + C)$. As a result, a value of confidence $C = 0$ does not affect the layout, while values of $1, 2, \ldots 9$ result in an intuitive two fold, three fold, … ten fold magnification of class separation. Note that in the KK algorithm the penalty for violating a given length constraint is inversely proportional to square of the desired edge length. In this way the contraction strategy manages to allocate more importance to the k-nearest neighbors edges. Note also that the contraction does not imply necessarily that single compact clusters will be enforced; if a class naturally decomposes in multiple groups, these will be individually contracted but will most likely remain separated, revealing that the user defined class assignment was too coarse.

**Ternary plots.** To better evaluate and visualize the dependencies between clusters we employ ternary or simplex plots (see Fig. 1h). A ternary plot graphically shows

in two dimensions the ratios of three variables as positions in an equilateral triangle under the constraint that they sum to a constant. In our case the three variables represent the probability that an instance belongs to one of the three classes under consideration. The Cartesian coordinates for a point representing the triple $(a, b, c)$ where $a = 100\%$ is located at $(0, 0)$, $b = 100\%$ is located at $(1, 0)$ and $c = 100\%$ at $\left( \frac{1}{2}, \frac{\sqrt{3}}{2} \right)$ can be readily computed as $\left( \frac{1}{2} \frac{2b+c}{a+b+c}, \frac{\sqrt{3}}{2} \frac{c}{a+b+c} \right)$. To compute the probability for each instance we solve a multi-class problem using the logistic regression[33] technique. In particular we employ the one-vs-rest (OvR) scheme where we solve as many binary classification problems as there are classes. For each binary problem we estimate the the probability that an instance $p_i \in \mathbb{R}^n$ belongs to class $y$ as:

$$P(Y_i = y | p_i) = \frac{e^{\beta p_i, y}}{1 + e^{\beta p_i}}$$

where the coefficient $\beta$ are estimated using the liblinear solver[34] and the probabilities are computed using a 5 fold cross-validation.

The ternary plots allow to visually gauge whether instances belonging to a given class can be clearly distinguished (the point cloud is concentrated mainly in the class vertex), or if the class definition is ambiguous or incorrect (no structure in the point cloud is visible) or if there is likely a transition from one class to the other via the third (a C shaped distribution in the point cloud can be detected). The underlying assumption here is that if the predictive model cannot discriminate whether an instance belongs to class $a$ or $b$ than the instance is likely to be in a transitional state between the two classes and is represented as a point located along the axis connecting $a$ and $b$. We define a confusion score to quantitatively represent this degree of "transition" from one class to the other via the third, as follows: for every instance we compute $P(Y_i = y | p_i)$, that is the probability of belonging to each one of the three classes under consideration; we average the score across instances belonging to the same class to obtain the confusion matrix $D$, a 3 by 3 matrix where $D_{ij}$ contains the average probability of classifying an instance belonging to class $i$ as being of class $j$; we symmetrize $D$ as $D' = \frac{DD^T}{2}$ and consider the cumulative error vector $E = \langle D'_{12}, D'_{13}, D'_{23} \rangle$; given the minimum element $E_m$ with $m = \arg \min E$ we define the score as $S = \frac{E_m}{\sum_{m' \neq m} E_{m'}}$ that is, as the ratio of the minimum multiclass error over the sum of the remaining two multiclass errors. A small score value indicates the presence of a transition. The intuition behind this choice is that a high multiclass errors between two classes indicates that there are intermediate instances that cannot be clearly classified as belonging to only one of the classes. Hence when we find that one pair of classes has a small error and the other two pairs have a high error we can infer the presence of a transition from one class to the other via the third.

**Comparison to Monocle2 and TSCAN.** Monocle 2[28] was run on non-normalized transcript counts with min_expr = 20 and num_cells_expressed ≥ 5. Cell types of the B cell, megakaryocyte, erythrocyte, and neutrophil/monocyte lineages were assigned based on expression of *Pax5*, *Pf4*, *Hba-a1*, and *Elane*, require a minimum expression of 5, 10, 20, and 50, respectively. Differentially expressed genes (qval < 0.01) were identified for each cell types and the top 100 genes for each cell type were selected for downstream analysis. Dimensions were reduced using DDRTree with max_components = 2, norm_method = log. For GraphDDP clusters obtained by RaceID3[27] run with mintotal = 10,000, minexpr = 17, outminc = 17, probthr = 1e-4, CGenes = ("Mki67","Pcna"), and default parameters otherwise, were used as input. GraphDDP was run with the parameters: --feature_selection --correlation_transformation --min_threshold = 5 -c 1 -k 3 -d 1 -l 3 -z 5 --random_state=1.

For the comparison with TSCAN, we downloaded the most recent package available on github (https://github.com/zji90/TSCAN, version 1.7.0) and used the provided R-script GUI with mainly default values except for the number of clusters. This implies that we used a log 2 transform as proposed by TSCAN (we tried without this transformation, but it did produce worse results). Furthermore, each cluster had to contain at least 5% of all genes. For the intestine case, this implied that we had to reduce the number of clusters to the optimal value proposed by TSCAN (6). For myeloid, we could use 20 as number of possible clusters. We then used the TSCAN built-in method to determine the pseudotime. The other possibility, Monocle, was not used as we have a comparison to the more recent Monocle2.

**Code availability.** The source code is available at: https://github.com/fabriziocosta/GraphEmbed. In addition we provide a Jupiter notebook[35] where the user can set all the parameters using sliding bars and observe interactively the effect on the layout.

## Data availability

All data is accessible in GEO with the following accession codes: (a) for the myeloid progenitors data[21], the accession code is GSE72857; (b) for he mouse multipotent hematopoietic progenitors[26], the accession code is GSE81682; (c) our intestinal data has the accession code GSE76408.

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

## Acknowledgements

This work was supported by the University of Freiburg and the Max Planck Society. Furthermore, we received support by the Deutsche Forschungsgemeinschaft (CRC992 Medep, TRR167 Neuromac (to R.B.) and GRK 2344 MeInBio (to D.G. and R.B.)), as well as the BIOSS Centre for Biological Signalling Studies (to R.B.).

## Author contributions

D.G. and R.B. conceived the study. F.C. designed and implemented the algorithm in collaboration with R.B. and D.G. D.G. and R.B. analyzed and interpreted the data. All authors wrote the manuscript.

## Additional information

**Competing interests:** The authors declare no competing interests.

