## [Peer Review File · Nature Communications]

Reviewers' comments:

Reviewer #1 (Remarks to the Author):

-I find incorporation of clusters information in visualization/finding differentiation paths between cell types a helpful approach. In order to show the added value of considering clusters information, I would like to ask for a simple comparison; how does a dimension reduction (e.g. by t-SNE with large perplexity or preferably diffusion maps to keep the global structure) look like when performed on the same set of features and normalizations as done in the preprocessing step?

-Does the algorithm allow refinement of cell's assignment to the clusters that is provided by the user, or does the use-provided clustering labels remain fixed? One can imagine occurrence of few mistakes in assignment of cells to clusters that is provided by the user.

-The method uses a force-directed approach like t-SNE which can give priority to preservation of local similarities and result in breaking of the global structures. How does the method guarantee preservation of global structures?

-As an additional test, can the method capture the lineage tree of LT-HSC, LMPP, CLP, GMP and PreMegE cell types from the sc-qPCR data published in "Characterization of transcriptional networks in blood stem and progenitor cells using high-throughput single cell gene expression analysis" Moignard. et al. Nat. Cell Biol. 2013?

-I appreciate investigation of local structures and relations in a different scale and resolution than the complete data is analyzed by using Ternary plots. However, shouldn't the Ternary plots be consistent with the edges displayed in the GraphDDP visualization? Is the Ternary repeating the same information as displayed by the edges? How reliable are the edges displayed by GraphDDP visualization of the data?

Reviewer #2 (Remarks to the Author):

This paper presents a visualization approach that aims to capture both discrete cluster and continuous trajectory relationships in single cell RNA-seq datasets. The use of supervised class labels and special edges to capture trajectory structure is somewhat interesting. However, I am not convinced that the method can give any unique biological insights that one couldn't also get from a previously developed trajectory construction approach or even a simpler strategy for connecting clusters.

Major Comments:

1. The paper states that "The recent StemID algorithm...is a first attempt of combining cluster information and trajectory inference." However, this is not really true. There are a number of previously published methods that use cluster information to build trajectories describing differentiation (or other processes). The authors should cite these papers and explain how the proposed visualization method extends or improves upon this previous work. Relevant methods include, for example, TSCAN, SCUBA, waterfall, ECLAIR, and slingshot. In short, it seems to me that the idea of identifying clusters and stitching them together into a trajectory is not a new idea, and there has been a lot of work along these lines already.

2. A crucial, related question is this: What specific biological insights can be drawn from this method but not previous methods? In particular, what does this visualization approach give you that you can't get from just running one of the trajectory approaches mentioned above, or even from clustering the data and connecting the clusters in a straightforward manner?

3. A key issue in reconciling the perspective of clusters vs. continuous trajectories involves determining, for a given dataset, whether a set of cells really are discretely separated or continuously joined. Does the proposed method provide any insight into this question? It strikes me that making a supervised visualization by forcing defined clusters to move apart just muddies the waters further. Also, pushing clusters further apart based on the "confidence" that one assigns to the clustering seems odd. What about a situation in which clusters correspond to quantiles along a continuous trajectory? Even though one can confidently identify the quantiles, a good visualization would ideally preserve the information that adjacent clusters are very close together. And it seems like such a situation is likely to arise frequently, since the method is intended to give insight into how clusters relate to trajectories.
4. I'm not convinced that the "k-shift edges" are necessary or helpful. How different would the visualization look if a simpler strategy were used to connect cells across clusters? The simplest strategy I can imagine is building a knn graph and allowing cells across clusters to be connected. It seems like this simple strategy should give a pretty decent force layout, and the visualization would also be more easily interpretable because the edges have a simple meaning. I also don't quite get the rationale behind the k-shift edges. On one hand you suggest that differentiation proceeds from high to low density, which I'm not sure is strictly true. But then you also say that "confluent differentiation would materialize between points of high densities", which seems to contradict the argument for connecting clusters of different densities. Also, I can't tell whether these k-shift edges are directed, although I assume they are not because there are no arrows in the visualizations. But the discussion about connecting high and low densities leads me to think that the edges are intended to go from high to low density.
5. What is the advantage of drawing the convex hull of each cluster? It certainly looks nice in the visualizations, but does this convey any information beyond what a simple color scheme would show?
6. The ternary plots in Fig. 3, which supposedly contrast a clear differentiation pathway and no differentiation pathway, look basically the same to me. It seems like a statistical test of some sort, rather than just a qualitative plot, would be helpful here.

Minor Comments:

1. The introduction mentions that PCA, MDS, and t-SNE have different biases. However, I'm not sure that the properties of PCA, MDS, and t-SNE stated here are correct. In particular, I've never heard that "PCA prefers to identify the two main directions of change" or "t-SNE has the propensity to segregate data points into a detached groups with relative arbitrary positions". If these descriptions are correct, you need to provide additional explanation or citations to papers that explain why these properties are true.
2. There are several minor typos:
Abstract: "combine both types on information"
The word "epithelial" is frequently spelled "epethelial" throughout the paper
"strategy menages to allocate" on p. 7
"allow to visually gauge weather" and "cannot discriminate weather" on p. 8

Answer to GraphDDP Reviewer Comments

We have thoroughly revised our manuscript according to the reviewers' comments and have added several validations and comparisons to existing methods. These changes are marked in red in the current manuscript.

Reviewer #1

1. *I find incorporation of clusters information in visualization/finding differentiation paths between cell types a helpful approach. In order to show the added value of considering clusters information, I would like to ask for a simple comparison; how does a dimension reduction (e.g. by t-SNE with large perplexity or preferably diffusion maps to keep the global structure) look like when performed on the same set of features and normalizations as done in the preprocessing step?*

- We performed the experiment of running t-SNE with a higher perplexity (new Suppl. Figure 4). In the myeloid data set, while a higher perplexity can ensure a connected global structure, it also blurs the separation of the different clusters, hindering the detection of pathways (see Supplementary Figure 4). The same holds for the intestine data (data not shown). Here is the t-SNE visualization for the myeloid data with a large perplexity P ($P=50$), where you can clearly see this effect (more details in the supplementary data).

2. *Does the algorithm allow refinement of cell's assignment to the clusters that is provided by the user, or does the user-provided clustering labels remain fixed? One*

can imagine occurrence of few mistakes in assignment of cells to clusters that is provided by the user.

- The clustering information provided by the user is fixed but it is considered only as a hint by the algorithm. It is only used to bias the visualization, i.e., as prior information. Specifically, it does not imply that the visualised clusters are going to be compact or convex. Outliers or wrong assignments can end up clustering together to form disconnected components, indicating that the original classification is probably too coarse and needs refining.
3. *The method uses a force-directed approach like t-SNE which can give priority to preservation of local similarities and result in breaking of the global structures. How does the method guarantee preservation of global structures?*
- The k-NN edges do indeed connect locally similar instances, however the k-shift edges form a tree by construction which in turn ensures the global connectedness of the structure. We have shown that the global layout becomes fragmented if you omit the k-shift edges (novel Supplementary Figures 5+6, see also answer to question 4 of Reviewer 2 who has asked about the importance of k-shift edges).
4. *As and additional test, can the method capture the lineage tree of LT-HSC, LMPP, CLP, GMP and PreMegE cell types from the sc-qPCR data published in “Characterization of transcriptional networks in blood stem and progenitor cells using high-throughput single cell gene expression analysis” Moignard. et al. Nat. Cell Biol. 2013?*
- We addressed this point by analyzing a more recent larger single-cell RNA-seq dataset from the same lab covering the entire pool of murine multipotent hematopoietic progenitors (Nestorowa et al. (2016) Blood) and included a novel Supplementary Figure 9 to present the results. Our analysis demonstrates that GraphDDP recovers the known architecture of the hematopoietic lineage tree based on the most recent findings both from single-cell RNA-seq and lineage tracing studies as summarized in a recent review (Laurenti and Gottgens (2018) Nature). This includes the continuous emergence of fate bias in the HSC compartment (Velten et al. (2017) Nature Cell Biology; Herman et al. (2018) Nature Methods), early divergence of megakaryocytes (Rodriguez-Fraticelli et al. (2018) Nature) and erythrocytes (Paul et al. (2015) Cell) as well as the existence of a lymphoid primed multipotent progenitor upstream of the CLP and the GMP. In contrast, Monocle2 (Qui et al. (2017) Nature Methods), a recently published state-of-the-art methods for lineage tree inference, could not recapitulate some of these key aspects. For instance, it did not place the CLP upstream of more mature B cell lineage populations and inferred a common progenitor of B cells and erythrocytes, while neutrophils/monocytes were predicted as an outgroup. This comparison demonstrates the excellent performance of GraphDDP for inferring complex differentiation hierarchies.
5. *I appreciate investigation of local structures and relations in a different scale and resolution than the complete data is analyzed by using Ternary plots. However, shouldn't the Ternary plots be consistent with the edges displayed in the GraphDDP visualization? Is the Ternary repeating the same information as displayed by the edges? How reliable are the edges displayed by GraphDDP visualization of the data?*

- We display k-shift edges to link neighbors belonging to a different class. The number of links between two classes gives a visual clue on the strength of the relationship between classes. The triangular plots on the other hand, show how consistently cell classes can be 'predicted'. The idea is to visualize, when considering three cell lines A,B,C, if, say, A and B contain intermediate members that cannot be clearly classified as belonging to only one of A or B. If the same happens for, say, B and C but not A and C, that is A and C are clearly and easily distinguishable, then we have an indication of a flow of the type A -> B -> C or C->B->A.
- To quantify this notion we have introduced a score computed from the confusion matrix as follows: score = ratio of minimum multiclass error over the sum of the remaining two multiclass errors. A small score value indicates the presence of a flow. The intuition behind this choice is that a high multiclass errors between two classes indicates that there are intermediate instances that cannot be clearly classified as belonging to only one of the classes. Hence when we find that one pair of classes has a small error and the other two pairs have a high error we can conclude that a flow is present. Now, in Figure 1, we have optimized the ternary representation (including the cells of all three classes displayed, colored according to the class) and in the Ternary Plots Section we have added the described score.

Reviewer #2

Reviewer #2 Major Comments

This paper presents a visualization approach that aims to capture both discrete cluster and continuous trajectory relationships in single cell RNA-seq datasets. The use of supervised class labels and special edges to capture trajectory structure is somewhat interesting. However, I am not convinced that the method can give any unique biological insights that one couldn't also get from a previously developed trajectory construction approach or even a simpler strategy for connecting clusters.

- We demonstrate the superior performance of GraphDDP in comparison to state-of-the-art methods by benchmarking on a published dataset of murine hematopoietic progenitors (Nestorowa et al. (2016) Blood). A large number of recently published single-cell RNA-seq and lineage-tracing-based studies has led to a revised ground truth model of the architecture of the hematopoietic lineage tree (reviewed in Laurenti and Gottgens (2018) Nature). GraphDDP clearly recovers the key aspects of this model (novel Supplementary Figure 9)

including continuous emergence of fate bias in the HSC compartment (Velten et al. (2017) Nature Cell Biology; Herman et al. (2018) Nature Methods), early divergence of megakaryocytes (Rodriguez-Fraticelli et al. (2018) Nature) and erythrocytes (Paul et al. (2015) Cell) as well as the existence of a lymphoid primed multipotent progenitor upstream of the CLP and the GMP. In contrast, Monocle 2 (Qui et al. (2017) Nature Methods), a recently published state-of-the-art methods for lineage tree inference could not recapitulate some of these key aspects. For instance, Monocle2 did not place the CLP upstream of more mature B cell lineage populations and inferred a common progenitor of B cells and erythrocytes, while neutrophils/monocytes were predicted as an outgroup. This comparison demonstrates the excellent performance of GraphDDP for inferring complex differentiation hierarchies compared to existing methods.

1. *The paper states that “The recent StemID algorithm...is a first attempt of combining cluster information and trajectory inference.” However, this is not really true. There are a number of previously published methods that use cluster information to build trajectories describing differentiation (or other processes). The authors should cite these papers and explain how the proposed visualization method extends or improves upon this previous work. Relevant methods include, for example, TSCAN, SCUBA, waterfall, ECLAIR, and slingshot. In short, it seems to me that the idea of identifying clusters and stitching them together into a trajectory is not a new idea, and there has been a lot of work along these lines already.*
 - In Supplementary Data we have included a comparison with TSCAN on the intestine and myeloid data (novel Supplementary Figures 7 and 8). The automatic clustering did not work quite well, which is expected, as the user-defined clustering also includes some manual curation. We also assessed whether the pseudotime shows the differentiation pathways detectable in the GraphDPP visualization. We investigated this by coloring our layout with grey-levels according to the defined pseudotime. For the intestine data, the pseudotime has more or less a vertical direction in our layout and all the cells in middle axis do not have a clear pseudotime ordering. For the myeloid data, it seems that the pseudotime is more pronounced within the clusters provided by the original publication (Paul et al. (2015) Cell) but does not show clear pathways:

2. *A crucial, related question is this: What specific biological insights can be drawn from this method but not previous methods? In particular, what does this visualization approach give you that you can't get from just running one of the trajectory approaches mentioned above, or even from clustering the data and connecting the clusters in a straightforward manner?*
 - As stated in the previous answers, GraphDPP could reconstruct cell hierarchies from scRNA-seq with given ground truth that could not be determined from trajectory approaches such as Monocle2 or TSCAN. Using a pure k-nearest neighbor approach (even in combination with the contraction enhancement) does not reveal clear pathways in many cases and leads to noisy visualizations, especially for smaller clusters. Using a straightforward approach to connect clusters (i.e. using only the k-NN edges) does not achieve satisfactory results because of the high amount of noise present in the profiles, which is why we propose to use k-shift edges that make use of a notion of averaging (i.e. density) to compensate for the noise.

3. *A key issue in reconciling the perspective of clusters vs. continuous trajectories involves determining, for a given dataset, whether a set of cells really are discretely separated or continuously joined. Does the proposed method provide any insight into this question? It strikes me that making a supervised visualization by forcing defined clusters to move apart just muddies the waters further. Also, pushing clusters further apart based on the "confidence" that one assigns to the clustering seems odd. What about a situation in which clusters correspond to quantiles along a continuous trajectory? Even though one can confidently identify the quantiles, a good*

visualization would ideally preserve the information that adjacent clusters are very close together. And it seems like such a situation is likely to arise frequently, since the method is intended to give insight into how clusters relate to trajectories.

- The key insight in the proposed approach is that it is possible to improve the quality of the embedding by exploiting prior knowledge given in input as a preferred clustering. If the provided cluster assignment is inconsistent the resulting layout will produce separate sub-groups since not all instances will be neighbors. At a coarser scale the layout of the groups will follow a trajectory via the k-shift-edges allowing to identify differentiation pathways (see for example the case of the myeloid data in Fig. 3). The confidence parameter is then used to obtain a higher visual contrast of the relative positioning of the various cell classes (see for example novel Supplementary Figure 6). Note that classes that are closer together (as the notion of quantiles proposed by the reviewer) will be placed near to each other as the preferred edge length for any edge type is proportional to the similarity between cell profiles.

4. *I'm not convinced that the "k-shift edges" are necessary or helpful. How different would the visualization look if a simpler strategy were used to connect cells across clusters? The simplest strategy I can imagine is building a knn graph and allowing cells across clusters to be connected. It seems like this simple strategy should give a pretty decent force layout, and the visualization would also be more easily interpretable because the edges have a simple meaning.*

- Employing only k-NN links (as it is used in the SPRING methods) does not guarantee a global tree structure. In the Supplementary Data (novel Supplementary Figures 5+6) we now report what happens when considering only k-NN edges. When k is low, then arbitrary disconnected components arise; moreover, when not using a class-specific contraction strategy, then clusters do overlap. And when k is high the resulting layout is noisy.
- One alternative would be to use a contraction strategy similar to the one proposed in our paper, i.e. contracting edges between instances of the same class. However this case results in fragmented representations, especially in the less populated cases, as can be seen here

- An alternative to the k-shift tree to guaranteed global connectedness would be to use the minimum spanning tree (MST). However MST are known to be fragile, that is, a single noisy instance can have a dramatic effect on the global layout as it can short circuit minimum paths. The k-shift tree is instead robust to outliers as it is based on the notion of *density* which is an averaged measure.
5. *I also don't quite get the rationale behind the k-shift edges. On one hand you suggest that differentiation proceeds from high to low density, which I'm not sure is strictly true. But then you also say that "confluent differentiation would materialize between points of high densities", which seems to contradict the argument for connecting clusters of different densities.*
- The k-shift links allow to connect density centers in an increasing order of density or closeness to a center of mass that should correspond to a more undifferentiated state. Please note that each clusters contains several density centers. Each denser location should correspond to sub populations of that cluster.

Also, I can't tell whether these k-shift edges are directed, although I assume they are not because there are no arrows in the visualizations. But the discussion about connecting high and low densities leads me to think that the edges are intended to go from high to low density.

- We do not explicitly consider the edge direction although it is easy to visualize a natural flow from the periphery of the layout towards the center.

What is the advantage of drawing the convex hull of each cluster? It certainly looks nice in the visualizations, but does this convey any information beyond what a simple color scheme would show?

- We just had the impression that it helps to detect all members of a class. The reason is that otherwise, unsimilar cluster members far from the cluster center could easily be overlooked. However, this is only a feature that can be switched off.

6. *The ternary plots in Fig. 3, which supposedly contrast a clear differentiation pathway and no differentiation pathway, look basically the same to me. It seems like a statistical test of some sort, rather than just a qualitative plot, would be helpful here.*
 - Thank you for the comment, we have now devised a scoring strategy to quantify the notion of flow between classes, please see answer to Reviewer 1 question 5.

Reviewer #2 Minor Comments

1. *The introduction mentions that PCA, MDS, and t-SNE have different biases. However, I'm not sure that the properties of PCA, MDS, and t-SNE stated here are correct. In particular, I've never heard that "PCA prefers to identify the two main directions of change" or "t-SNE has the propensity to segregate data points into a detached groups with relative arbitrary positions". If these descriptions are correct, you need to provide additional explanation or citations to papers that explain why these properties are true.*
 - We have reformulated and weakened our statement.
2. *There are several minor typos:*

Abstract: "combine both types on information"

The word "epithelial" is frequently spelled "epethelial" throughout the Paper

"strategy menages to allocate" on p. 7

"allow to visually gauge weather" and "cannot discriminate weather" on p. 8

 - Many thanks, we have corrected the typos.

REVIEWERS' COMMENTS:

Reviewer #1 (Remarks to the Author):

The authors have addressed my questions.

I recommend few discussion lines about the impact of user-defined cell labels and what could go wrong with poor quality user-defined cell labels (e.g. curse of dimensionality problems for densest nearest neighbours identification if too few classes are defined?).

The approach is usefull and demonstrates how pieces of information about cell differentiation hierarchies that would be missed by fully unsupervised statistical approaches, can be explored via few (supervised) interventions.

Reviewer #2 (Remarks to the Author):

The authors have satisfactorily addressed my comments. I better understand the importance of the k-shift edges, and this seems to be one of the key insights that distinguishes this method from previous approaches. The comparisons with TSCAN, SPRING, and Monocle2 demonstrate some scenarios in which the new strategy provides certain advantages.

Answers to Reviewers Comments.

Reviewer #1

The authors have addressed my questions. I recommend few discussion lines about the impact of user-defined cell labels and what could go wrong with poor quality user-defined cell labels (e.g. curse of dimensionality problems for densest nearest neighbours identification if too few classes are defined?). The approach is usefull and demonstrates how pieces of information about cell differentiation hierarchies that would be missed by fully unsupervised statistical approaches, can be explored via few (supervised) interventions.

We would like to thank the reviewer for his comments. Concerning the impact of user-defined cell labels and what could go wrong, we have the feeling that we have made clear enough in the paper that the cell labels are a critical prior information. Nevertheless, in the paragraph entitled "User defined confidence strength in class assignment", where we have added the following sentence:

Please note that the user-defined clustering is a critical prior information, and wrong layouts are generated if wrong or low quality class assignments are provided.

Reviewer #2

The authors have satisfactorily addressed my comments. I better understand the importance of the k-shift edges, and this seems to be one of the key insights that distinguishes this method from previous approaches. The comparisons with TSCAN, SPRING, and Monocle2 demonstrate some scenarios in which the new strategy provides certain advantages.

We would like to thank the reviewer for his comments.